# This is *not* a Dataset:
# A Large Negation Benchmark to Challenge Large Language Models

**Iker García-Ferrero**[1] , **Begoña Altuna**[1] , **Javier Álvez**[2]
**Itziar Gonzalez-Dios**[1] , **German Rigau**[1]

[1] HiTZ Center - Ixa, University of the Basque Country UPV/EHU
[2] LoRea Group, University of the Basque Country UPV/EHU
{iker.garciaf,begona.altuna,javier.alvez}@ehu.eus
{itziar.gonzalezd,german.rigau}@ehu.eus

## Abstract

Although large language models (LLMs) have apparently acquired a certain level of grammatical knowledge and the ability to make generalizations, they fail to interpret negation, a crucial step in Natural Language Processing. We try to clarify the reasons for the sub-optimal performance of LLMs understanding negation. We introduce a large semi-automatically generated dataset of circa 400,000 descriptive sentences about commonsense knowledge that can be true or false in which negation is present in about 2/3 of the corpus in different forms. We have used our dataset with the largest available open LLMs in a zero-shot approach to grasp their generalization and inference capability and we have also fine-tuned some of the models to assess whether the understanding of negation can be trained. Our findings show that, while LLMs are proficient at classifying affirmative sentences, they struggle with negative sentences and lack a deep understanding of negation, often relying on superficial cues. Although fine-tuning the models on negative sentences improves their performance, the lack of generalization in handling negation is persistent, highlighting the ongoing challenges of LLMs regarding negation understanding and generalization. The dataset and code are publicly available: https://github.com/hitz-zentroa/This-is-not-a-Dataset

## 1 Introduction

Large Language Models (LLMs) currently offer state of the art performance in many Natural Language Processing (NLP) tasks. Apparently, they have acquired the ability to capture syntactic (Baroni, 2020) and semantic (Furrer et al., 2021) abstractions. However, recent experiments (Kassner and Schütze, 2020; Hossain et al., 2020; Truong et al., 2022) have proven that LLMs fail at interpreting contexts in which understanding negation is required.

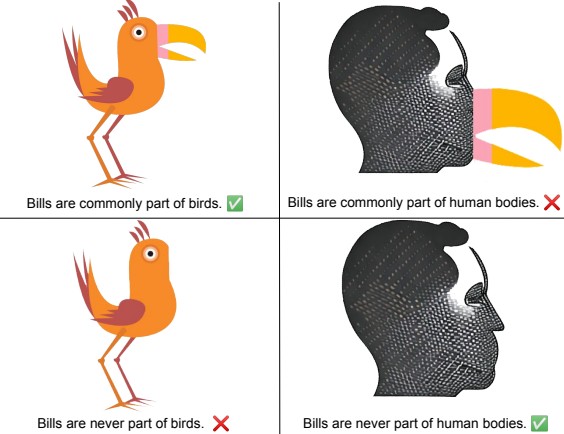

Figure 1: Affirmative and negative sentences in the dataset.

The presence of negation in a sentence reverts the polarity of the proposition it represents, and thus affects its truth and factuality values. See how the adverb "never" changes the truth value of the sentences in Figure 1. As a consequence, understanding negation correctly is crucial for all NLP tasks. Moreover, understanding negation should help LLMs to grasp how things happen in reality, boosting NLP tasks that involve commonsense, causality, entailment and world knowledge.

The reasons for the lower capabilities of LLMs dealing with negation remain largely unclear, although some point out at the under-representation of negation in corpora (Hossain et al., 2022). In this work, we present a corpus in which negation is present in around two thirds of the sentences in different forms. Taking advantage of the relations in WordNet (Fellbaum, 1998), we have generated a set of patterns to create descriptive sentences that work as truth and falsity tests which are then used together with a list of prompts to measure the sentence understanding of the different LLMs.

The dataset has been used in a series of experiments to test its quality and coherence. First, we assess the quality of the sentences by human anno-

tators. Then, to grasp its capacity of generalization and inference, we have used our dataset to test different configurations of LLMs available in a zero-shot approach. We have also fine-tuned some of these models to assess whether the understanding of negation can be learnt. Our initial hypothesis is that if the dataset is coherently and robustly built we will be able to learn how LLMs deal with negation.

The contributions of this paper are: i) We introduce the largest negation probing dataset. This dataset includes affirmative and negative sentences with and without distractors, incorporating multiple types of relations and negations. ii) We evaluate a comprehensive set of open LLMs using our dataset in both zero-shot and fine-tuning scenarios. iii) Our findings demonstrate that current LLMs, whether in zero-shot settings or after fine-tuning with examples from our dataset, possess a profound understanding of the truthfulness of affirmative sentences. However, when confronted with negation, these models heavily rely on superficial cues instead of effectively generalizing negation.

## 2 Background

### 2.1 Related Works

Negation is a core operator in logic and in the structuring of the information in text and it has long been studied for its relevance in natural language understanding. In the last two decades, works on the analysis and processing of negation have multiplied. In the pre-generative-model era, most works centered on negation detection (Chapman et al., 2001; Vilares et al., 2015) and profiling (Morante and Daelemans, 2012), so the extracted negation information could be used in downstream tasks.

With the booming of deep-learning architectures that were based on abstract neural representations of texts, the paradigm shifted and negation was processed as the rest of the elements appearing in text. It was soon noticed that systems struggled to correctly process the information when negation was involved. Such is the case of negation in machine translation (Bentivogli et al., 2016; Tang et al., 2021), information extraction (Grivas et al., 2020) and sentiment analysis (Barnes et al., 2021) among others.

It has not been long since the scholar community started to analyse the reasons for the lack of capability of correctly processing negation. For example, Jumelet and Hupkes (2018) analysed the

negation licensing strategies to measure neural language model ability to correctly process them.

Chen et al. (2023) assess the ability of LLMs to handle negative commonsense knowledge. Since most available information exists in a positive and affirmative form, LLMs fail at dealing with world knowledge when it is presented in a negative form. They propose a two-task assessment in which LLMs need to i) answer *yes* or *no* to world knowledge questions and ii) generate commonsense compelling sentences from related keywords. Some recent research has been directed to building knowledge bases in which negative commonsense is stored (Arnaout et al., 2022), in order to be reused for commonsense reasoning.

### 2.2 Negation in English

Negation in language is the representation of the logical operation in which a the truth value of a proposition is inverted. It is commonly expressed by a restricted list of negative adverbs (e.g. *no*, *never*), pronouns, determiners or prefixes, that appear in different contexts in the sentence. Pullum et al. (2002) offer a four axe classification of negation types from which we will focus on these:

- **Verbal vs. non-verbal**: the verbal negation marker is associated with the verb and directly affects it, while the non-verbal couples with objects and adjuncts.

- **Analytic vs. synthetic**: analytic negation is represented by markers that only convey negation. Synthetic negation markers, instead, may have additional syntactic function (e.g. *nothing* and *none* might also be subjects or objects).

- **Clausal vs. sub-clausal**: clausal negation negates the whole clause that includes it, and sub-clausal negation only affects a part of the clause.

In Table 1 we present the different types of negations considered in our work.

## 3 Dataset

### 3.1 Dataset construction

Our benchmark compiles 381,300 artificially generated sentences in standard English. The sentences, in a definition style (e.g. X is Y, X is part of Y), have been created based on the knowledge from WordNet and related resources.

| Negation type | Example |
|---|---|
| Verbal | Agreement is *not* an appropriate synonym of disagreement in any context. |
| Non-verbal | In *no* context lectures may be part of courses. |
| Analytic | *No* theft is a small replica of a person. |
| Synthetic | Mirror is *never* an appropriate hyponym of reduction. |
| Clausal | Kissing is *not* commonly done by engineers. |
| Sub-clausal | Bricks are made of clay in *no* context |

Table 1: Examples of the types of negation.

| Pattern | Relation | Templates | Triples | Sentences |
|---|---|---|---|---|
| #01 | Synonymy | 21 | 2,996 | 281,624 |
| #02 | Antonymy | 21 | 58 | 8,178 |
| #03 | Synonymy | 24 | 14 | 2,436 |
| #04 | Antonymy | 24 | 58 | 10,092 |
| #05 | Hypernymy | 24 | 634 | 60,864 |
| #06 | Part | 16 | 199 | 11,940 |
| #07 | Substance | 15 | 21 | 1,176 |
| #08 | Member | 17 | 11 | 682 |
| #09 | Agent | 2 | 60 | 240 |
| #10 | Instrument | 7 | 9 | 468 |
| #11 | Result | 27 | 40 | 3,600 |
| Total | - | - | - | 381,300 |

Table 2: Distribution of sentences by pattern.

The sentences in our dataset are obtained by means of patterns. Each of the 11 patterns (#01–#11) is designed for a particular relation and includes several different templates of two types: *affirmative* templates, which are free of negation; *negative* templates, which include one of the types of negations described in Table 1. Using triples on the corresponding relation, these templates are used to create sentences by instantiation. Since each synset may include more than one word form in Core WordNet and the proposed templates include optional and alternative parts, we obtain several sentences from each couple of template and triple. The controlled application of the different templates enables us to determine the truth-value of the resulting sentences. In Appendix D, we describe each pattern in detail.

More specifically, we focus on the WordNet relations *synonymy*, *hypernymy*, *antonymy*, *meronymy* (*part*, *member* and *substance*) and the semantic roles *agent*, *instrument* and *result* provided by *Morphosemantic Links* (Fellbaum et al., 2009). Among the nouns and verbs compiled in WordNet, we concentrate exclusively on the ones provided by *Core*

WordNet (Boyd-Graber et al., 2006), which is a list of the most frequently used word senses that includes 3,299 nouns and 1,000 verbs. In this way, we discard words that are less commonly used. Furthermore, we exclude the triples on synonymy[1] and hyponymy that relate *Basic Level Concepts* (BLCs) (Izquierdo et al., 2007), which may result too general, and we use the mapping from WordNet to *EuroWordNet Top Ontology* (TCO) to ignore the triples on the *member* meronymy relation and the *agent* semantic role where the noun synsets are not referring to animals or persons.

Since WordNet and the considered related resources only provide true knowledge —that is, all the triples and mappings describe real relations and connections— we automatically obtain false knowledge from WordNet triples using *distractors*, which are randomly selected words that replace the word senses of a synset.[2] That is, given a WordNet triple that relates two synsets, from Core WordNet we select a distractor to replace the word senses of one of the synsets and obtain a *distracting triple*. Apart from BLCs, for the selection of suitable distractors we consider the *lexicographer files* provided by WordNet, which are 45 syntactic category and logical groupings of word senses, and WordNet *Domains* (Bentivogli et al., 2004), which consist of a hierarchy of 164 labels that characterize knowledge areas and to which each synset is connected. In Appendix C, we provide more details about the selection of distractors.

Next, we illustrate the process of constructing our dataset. In Pattern #06, we have included the following positive and negative templates that state semantic correspondences between parts and wholes on the basis of triples of the form $\langle part, noun_1, noun_2 \rangle$:

$\langle noun_1+(e)s \rangle$ [ are commonly | may be ] part of $\langle noun_2+(e)s \rangle$.

$\langle noun_1+(e)s \rangle$ are never part of $\langle noun_2+(e)s \rangle$.

The positive template[3] yields true sentences when instantiated with true knowledge (i.e. WordNet triples), while we get false sentences using distracting triples. On the contrary, the negative one yields sentences with the opposite truth-value. For example, given the WordNet triple

---

[1] Synonymy triples are obtained by reflexivity.

[2] In Patterns #01 and #02, distractors are synsets when using glosses.

[3] The expressions enclosed in square brackets are alternative.

$\langle part, bill_n^{10}, bird_n^1 \rangle$, we select *human body* as distractor for $bird_n^1$ and get the distracting triple $\langle part, bill_n^{10}, human\ body \rangle$. Then, by instantiating the positive template using these two triples, we get the sentences in the first row of Figure 1 and also:

> "*Bills may be part of birds.*"

The two sentences about *birds* (i.e. resulting from the WordNet triple) are labelled with *True*, while the sentence about *human bodies* (that is, obtained from the distracting triple) is labelled with *False*. Likewise, using the same two triples for the instantiation of the negative template, we get the sentences in the second row of Figure 1, which are respectively labelled with *False* and *True*.

In Table 2, we sum up some figures about the proposed dataset. For each pattern (first column), we provide the corresponding WordNet relation and the number defined templates, applied WordNet triples and obtained sentences respectively. It is worth noting that Patterns #01–#04 include both false positive sentences and true negative sentences obtained from synonymy and antonymy WordNet triples by means of a dual application of templates. Furthermore, in the case of antonymy, the truth-value of the resulting sentences does not depend on whether templates are instantiated using WordNet or distracting triples. As a consequence, instantiating a template using a WordNet and a distracting triple yields sentences with opposite truth-value except for Patterns #02 and #04, where all sentences resulting from the same template have the same truth-value. For example, given the antonym triple $\langle ant, expenditure_n^1, income_n^1 \rangle$, we select *wood* as distractor for $expenditure_n^1$ (see Appendix C for details) and obtain the distracting triple $\langle ant, wood, income_n^1 \rangle$. Using these two templates, we instantiate the following negative template included in Pattern #04

> $\langle noun_1 \rangle$ and $\langle noun_2 \rangle$ are the same thing in no context.

and obtain two sentences:

> "*Expenditure and income are the same thing in no context.*"

> "*Wood and income are the same thing in no context.*"

Both sentences are true.

| % | A tester | B tester | A ∩ B | A ∪ B |
|---|---|---|---|---|
| T/F prediction | 90.9 | 89.1 | 87.27 | 96.36 |
| Comprehensibility | 91.82 | 100 | 91.82 | 100 |
| Grammaticality | 83.64 | 96.82 | 83.64 | 96.82 |
| Plausibility | 20.45 | 45.91 | 19.55 | 46.82 |

Table 3: Human evaluation of the quality of the sentences in the dataset.

## 3.2 Dataset quality assessment

Human Evaluation addresses the validation of the generation process and the different templates used, that is to say, whether the sentences in the dataset are grammatical and that overall represent true and false knowledge as expected. To prove the linguistic quality of the dataset and that the predictions extracted from WordNet reflect the reality, two native speakers of English were required to assess a randomly selected sample of 220 sentences from our dataset. Evaluators were required to answer four questions for each sentence: i) Is the sentence true or false?, ii) is the sentence grammatically correct?, iii) is the sentence understandable? and iv) is the sentence plausible and might be produced by a speaker?

The answers to these questions have been summarised in Table 3. For the true and false predictions, we have compared the testers' answers with the predictions we generated from the WordNet relations. Circa 90% of the predictions match with the human testers' answers. For the quality of the test sentences, the results show that the sentences in the dataset are mostly comprehensible to humans even if not all are fully acceptable in English or they are not likely to be uttered by English speakers (low plausibility). Namely, we have detected problems with uncountable nouns (1) and lexical selection (2). Nonetheless, low plausibility might be an interesting asset for our experiments as employing non-frequent sentences may help to reduce the effect of the reliance on lexical co-occurrences models have.

(1) "*A letter is commonly part of a mail.*"
(2) "*Officers are not members of laws in any context.*"

In what refers the quality of the knowledge encoded in the dataset, we have observed that over 98% of the sentences with distractors in the human test set represent actual knowledge. We can thus consider that the distractor selection mechanism is

robust enough.

# 4 Experimental Setup

In this section, we define the evaluation protocol we use to measure the performance of Language Learning Models (LLMs) on our dataset.

## 4.1 Models

We evaluate a diverse set of LLMs, ranging in size from 7 billion parameters up to 65 billion parameters. Our evaluation includes Foundation Models, along with versions that have undergone additional instruction-tuning and/or have been fine-tuned for conversation. We do not consider closed models where the data used for pretraining or even the model architecture is unknown, as drawing meaningful conclusions for such systems is not possible. We evaluate the following models: the 12 billion parameter **T5** (Raffel et al., 2020) encoder-decoder language model, as well as FLAN-T5 (Chung et al., 2022), an enhanced version of T5 that has been fine-tuned in a mixture of tasks; **LLaMA** (Touvron et al., 2023) decoder-only language models with parameter sizes ranging from 7 billion to 65 billion; LLaMA models that have been fine-tuned for specific tasks, including Vicuna v1.1 (Chiang et al., 2023), which has undergone additional fine-tuning as a chat-assistant, and WizardLM (Xu et al., 2023), which has been fine-tuned for following instructions; **Pythia** (Biderman et al., 2023) decoder-only 12 billion parameter model; the instruction-tuning model **Dolly** (Conover et al., 2023); and finally we evaluate **Falcon** (Almazrouei et al., 2023) 7 and 40 billion parameter models which are decoder-only models including the instruction-following fine-tuned versions. We also evaluate other open LLMs; the full model list can be found in Appendix A.

## 4.2 Task Formulation

We evaluate each sentence in the dataset individually as a binary task in which the model must generate either *True* or *False* tokens. Following Scheurer et al. (2023), given the prompt $pt$ we compute the answer $A$ as follows:

$$A = \begin{cases} True & if \frac{p(True|pt)}{p(True|pt)+p(False|pt)} > 0.5 \\ \\ False & otherwise \end{cases}$$

We use the following prompt as input for the models:

*Is the following statement True or False?*
*{sentence}.*

We found that models that have undergone a fine-tuning for conversation tend to generate an explanation instead of answering True or or False. We use a slightly modified prompt that improves the results: *Is the following statement True or False? Answer only True or False. {sentence}*. Models that have been fine-tuned as dialogue systems utilize different prompts to represent a conversation, such as using markers like "<bot>" and "<human>", or custom system initial prompts. In order to accommodate these models, we format the input according to the recommendations provided by the authors. Implementation details of fine-tuning and inference are available in Appendix B.

## 4.3 Metrics

In our dataset, we utilize two primary metrics for evaluating LLMs:

**Accuracy** This metric is computed using the formula $acc = (TP+TN)/(TP+TN+FP+FN)$. We evaluate the overall accuracy at the sentence level for all the sentences in our dataset. Additionally, we analyze the overall accuracy of different sentence types: Accuracy in Affirmative sentences, Negative sentences, Affirmative sentences with a distractor and Negative sentences that include a distractor.

**Coherence** This metric aims to decouple the real-world and commonsense knowledge of the model from the understanding of negative sentences. We compute two coherence scores: one for the sentences without distractors ("Bills are commonly part of birds" and "Bills are never part of birds") and another for the sentences with distractors ("Bills are commonly part of human bodies." and "Bills are never part of human bodies."). Answers are deemed coherent if the affirmative and negative sentences have opposite labels, regardless of whether the answer is correct or incorrect. However, if the model predicts the same label for both the affirmative and negative sentences, we consider the answer incoherent. To illustrate this metric, consider the sentence pair "Bills are commonly part of birds" and "Bills are never part of birds". Both the answers "True/False" and "False/True" are considered coherent, whereas "True/True" and "False/False" are incoherent.

Moreover, we calculate the overall coherence:

| Model name | Model Type | Coherence | | | Accuracy | | | | |
|---|---|---|---|---|---|---|---|---|---|
| | | | | | | W/o Distractor | | W/ Distractor | |
| | | All | W/o Distractor | W/ Distractor | All | Affirmation | Negation | Affirmation | Negation |
| Random | | 0.5 | 0.9 | 0.8 | 50.0 | 50.2 | 50.1 | 50.0 | 49.9 |
| LLaMA13B | Foundation | 0.0 | 0.2 | 0.3 | 50.1 | 85.8 | 12.2 | 10.6 | 90.2 |
| LLaMA30B | Foundation | 0.1 | 0.3 | 0.2 | 52.4 | 84.7 | 29.5 | 30.2 | 68.8 |
| LLaMA65B | Foundation | 0.0 | 0.0 | 0.0 | 50.3 | 96.3 | 3.1 | 1.3 | 99.3 |
| Vicuna13B | Dialogue | 0.2 | 8.8 | 0.6 | 57.8 | 83.1 | 85.1 | 78.0 | 2.6 |
| WizardLM30B | Instruction | 0.0 | 6.0 | 0.1 | 57.3 | 53.6 | 95.7 | 88.8 | 2.0 |
| Pythia12B | Foundation | 0.0 | 0.1 | 0.0 | 50.1 | 93.8 | 15.2 | 4.0 | 86.7 |
| Dolly12B | Instruction | 0.0 | 0.3 | 0.2 | 50.2 | 72.0 | 73.3 | 33.4 | 25.1 |
| T5-xxl | Foundation | 0.0 | 0.0 | 0.0 | 50.3 | 96.6 | 2.8 | 0.4 | 99.8 |
| Flan-T5-xxl | Instruction | 0.9 | 46.4 | 1.2 | 66.1 | 86.1 | 96.1 | 94.6 | 6.2 |
| Falcon40b | Foundation | 0.1 | 0.1 | 0.2 | 49.7 | 90.9 | 13.9 | 11.6 | 83.3 |
| Falcon40b-instruct | Instruction | 0.1 | 1.5 | 0.2 | 54.7 | 64.3 | 76.8 | 71.4 | 16.6 |

Table 4: Zero-shot performance of various LLMs in our dataset. The best results are highlighted in **bold**, and scores that surpass the Random baseline accuracy are underlined.

this happens when all the statements with and without distractors are coherent and correctly or all incorrectly classified. Referring to the example in Figure 1, we would deem the set of statements as overall coherent if the sentences with and without distractors are coherent and all the answers are either correct or all of them are incorrect. In the case of antonymy relations (Patterns #02 and #04), both the distractor-carrying and non distractor-carrying sentences carry the same label, so we evaluate the overall coherence accordingly. It is important to note that, for the sake of simplicity, the example only contains two sentences, but coherence is actually determined at the triple level. Triples can comprise between 2 to 27 templates. So, for a triple to be deemed coherent, responses to all the templates within it must be coherent. Therefore, this is a very challenging metric.

By examining coherence in these contexts, we gain insights into the models' ability to understand the negation, even if the models do not have the real-world knowledge to correctly label the sentences.

## 5 Do LLMs understand negation?

In this section, we assess the performance of the LLMs in section 4.1 in our dataset. The evaluation is conducted in a zero-shot setting, meaning that we evaluate the models without any fine-tuning. The results of this evaluation are presented in Table 4. Foundation models, which are trained on large amounts of unlabeled data, demonstrate an *All True* behavior. They accurately label as *True* the majority of affirmative sentences and negative

| | W/o Distractor | |
|---|---|---|
| Flan-T5-xxl | Affirmation | Negation |
| #01 Synonymy | 91.19 | 98.04 |
| #02 Antonymy | 96.36 | 25.62 |
| #03 Synonymy | 49.76 | 98.47 |
| #04 Antonymy | 82.07 | 21.92 |
| Vicuna13B | | |
| #01 Synonymy | 88.69 | 84.88 |
| #02 Antonymy | 71.65 | 4.64 |
| #03 Synonymy | 57.86 | 90.05 |
| #04 Antonymy | 75.8 | 12.81 |

Table 5: Accuracy of Flan-T5-xxl and Vicuna13B in the Synonymy and Antonymy patterns. We evaluate the models in affirmative and negative sentences without distractors. Scores that surpass the Random baseline are indicated with underline.

sentences with a distractor, which are *True* with the exception of the *Antonymy* patterns, that form approximately 5% of the total sentences. However, these models struggle to classify negative sentences and affirmative sentences with opposite labels. Their performance in these falls significantly below the random baseline exhibiting a total lack of coherence by the models.

Models that have undergone dialogue or instruction tuning, particularly Vicuna and FlanT5, demonstrate higher accuracy, instead. These models achieve a very high accuracy in sentences without a distractor. Specifically, Flan-T5 shows coherent answers for 46% of the triples. It is to be noted that this is a challenging metric, as a triple may be use to build up to 27 templates, and all of them must be coherent for the triple to be considered coherent.

| Model name | Coherence | | | Accuracy | | | | |
|---|---|---|---|---|---|---|---|---|
| | | | | All | W/o Distractor | | W/ Distractor | |
| | All | W/o Distractor | W/ Distractor | | Affirmation | Negation | Affirmation | Negation |
| Flan-T5-xxl | 51.8 | 55.4 | 92.9 | 94.1 | **96.5** | 86.7 | 96.1 | **98.0** |
| Vicuna13B | **81.2** | **86.4** | **94.2** | 95.7 | 92.7 | **94.4** | 98.1 | 97.2 |

Table 6: Performance of Vicuna13B and Flan-T5-xxl after fine-tuning in our dataset. The best results are highlighted in **bold**, and scores that surpass the Random baseline accuracy are underlined.

However, these models fail to correctly label negative sentences with a distractor. We further analyze the performance of Flan-T5 and Vicuna in negative sentences, focusing on the Synonymy and Antonymy patterns. In Pattern #01 and #02, as well as Pattern #03 and #04, the templates are opposite to each other, as explained in Subsection 3.1. Table 5 presents the performance of Flan-T5 and Vicuna in handling these patterns. Interestingly, both models achieve good results in negative sentences from the Synonymy patterns (labeled as *False*) but struggle with the negative sentences from the Antonymy patterns (labeled as *True*). This, along with their poor performance in negative sentences with a distractor (which are expected to be *True*, but models predict the label *False*), confirms that the models are heavily biased to always predict the label *False* in the presence of negation, regardless of the actual meaning of the sentence. This behavior suggests that the models lack a deep understanding of negation, and that they tend to rely on superficial cues rather than comprehending the true meaning conveyed by the negative sentences.

Despite the poor performance of the models in negative sentences, it is important to note that they demonstrate the ability to correctly label affirmative sentences, both with and without distractors. This demonstrates that the models have a deep understanding of truth and falsehood. Models' struggles primarily result from the presence of negation rather than a lack of comprehension or real-world knowledge.

## 6 Exposure to negation does not solve the problem

Understanding whether LLMs would understand negation if a sufficient number of negative sentences were present in the pretraining corpora is crucial for improving their reasoning capabilities and addressing the limitations associated with negative knowledge. However, due to the lack of suf-

ficiently large datasets containing negative knowledge, this hypothesis has not been extensively explored. In contrast, our dataset is substantial enough to be split into training, development, and test sets. To investigate whether LLMs can learn to reason over negative knowledge given enough negated data, we split the dataset at the triple level, ensuring that all sentences within a triple are assigned to the same split to ensure no data contamination. Our training dataset consists of 268,505 sentences from 2,876 triples, the development dataset includes 2,514 sentences from 244 triples, and the test dataset contains 90,281 sentences from 980 triples.

We finetune Flan-T5 and Vicuna on our dataset; the results are listed in Table 6. The impact of fine-tuning is remarkable, as it completely transforms the models' performance compared to their zero-shot counterparts. Both Flan-T5 and Vicuna exhibit higher accuracy than human annotators and achieve a notably high level of coherence. However, are the models truly learning about negation, or are they just exploiting patterns in the data? We conduct experiments to asses this.

First, we train Vicuna, the best performing model, using varying amounts and types of negative knowledge. We conduct separate fine-tuning experiments using all the affirmative sentences and all the negative sentences from the dataset, resulting in two distinct models. The results of this training are presented in Table 7. Training the model exclusively with affirmative sentences yields a high accuracy in the affirmative test sentences, but it labels incorrectly nearly all the negative sentences. Conversely, when trained solely with negative sentences, the model deals successfully with the negative sentences but struggles with the affirmative sentences. Despite being exposed to extensive real-world knowledge from WordNet, the models exhibit a significant failure in comprehending negation. They consistently overlook the presence of it

|  | Verbal | Non-Verbal | Analytic | synthetic | clausal | subclausal | Affirmation |
|---|---|---|---|---|---|---|---|
| All | 96.0 | 95.7 | **95.8** | 96.0 | 96.0 | 95.7 | 95.5 |
| All Affirmations | 6.2 | 6.8 | 6.8 | 5.9 | 6.1 | 6.8 | 95.7 |
| All Negated | **96.1** | **95.8** | **95.8** | **96.3** | **96.1** | **95.8** | 4.5 |
| Affirmations + Verbal | 95.5 | 79.5 | 81.9 | 95.6 | 95.5 | 79.5 | 95.4 |
| Affirmations + Non-Verbal | 94.9 | 95.6 | 95.2 | 96.1 | 94.9 | 95.6 | 95.8 |
| Affirmations + Analytic | 96.1 | 95.6 | 95.6 | 96.1 | 96.0 | 95.6 | 95.9 |
| Affirmations + synthetic | 94.8 | 44.5 | 51.8 | 96.0 | 94.9 | 44.5 | 95.6 |
| Affirmations + clausal | 95.8 | 34.6 | 43.9 | 96.0 | **96.1** | 34.6 | 95.8 |
| Affirmations + subclausal | 95.1 | 95.7 | 95.3 | 96.2 | 95.1 | 95.7 | **96.0** |

Table 7: Accuracy of Vicuna13B after fine-tuning with different types and amount of negative knowledge. The best results are highlighted in **bold**, and scores that surpass the Random baseline accuracy are indicated with underline.

and generate identical outputs for both affirmative and negative sentences. We also fine-tune models using various combinations of affirmative sentences and different types of negations. We observe that models trained with synthetic and clausal negations struggle to accurately classify non-verbal, analytic, and sub-clausal sentences. This suggests that while the models show proficiency in understanding and reasoning with certain types of negations, they face challenges in comprehending and correctly responding to other forms of negations that they have not seen in the fine-tuning step.

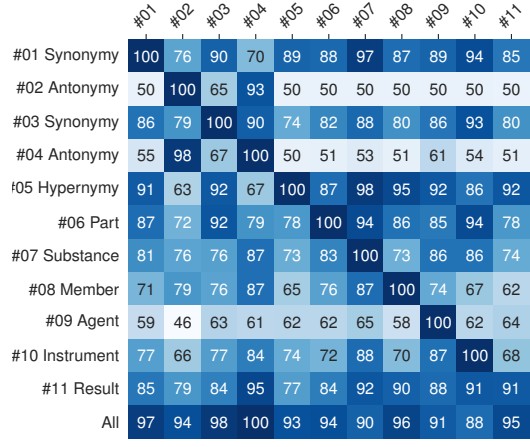

Figure 2: Evaluation of Vicuna13B accuracy when trained on one pattern (rows) and evaluated on the others (columns).

We also fine-tune Vicuna13B with each of the 11 patterns in our dataset independently, and we evaluate its performance on the other patterns. Figure 2 shows the overall accuracy scores. The results reveal that training the model with one pattern does not facilitate any successful generalization across all other patterns. Notably, as discussed in Section 3.1, the labels for affirmative and negative sentences from the Antonymy patterns are opposite

to those from the remaining patterns. The failure of models trained in other patterns to label the Antonymy patterns, as well as the failure of models trained in the Antonymy patterns to label other patterns, suggest that the models are relying on repetitive data structures that are not transferable to different patterns, rather than truly understanding the concept of negation. While exposure to negation may contribute to achieving favorable results within a specific dataset, it does not lead to a generalization on negation by the models. Negation continues to pose a significant challenge in the field of Natural Language Processing and remains an unsolved problem, requiring further research and development.

## 7 Conclusion

Current LLMs are typically trained using next token or mask token prediction objectives, which have proven effective for various NLP tasks. However, it remains an open issue understanding how certainly a model models negation. Negation tokens, which intermittently appear in sentences, hold little predictive importance for other tokens in the sentence. As a result, there is limited negation signal during language modeling training. Previous research has touched upon this issue but was limited by small manually generated datasets. In contrast, our study introduces the largest dataset to date comprising negative sentences. This comprehensive dataset includes affirmative and negative sentences with and without distractors, incorporating multiple types of relations and negations, which help to encode the underlying mechanisms for negation understanding. Through our analysis, we reveal that current LLMs, both in zero-shot settings and when fine-tuned with examples from our dataset, exhibit a profound understanding of

the truthfulness of affirmative sentences. However, when it comes to negation, these models heavily rely on superficial cues instead of generalizing negation and these superficial cues are not transferable across different negative sentences.

Negation remains a persistent and unsolved challenge in the field of NLP, demanding further research to develop systems capable of effectively handling it. Our dataset holds the potential to significantly contribute towards achieving this objective. In our future work, we plan to explore advanced reasoning paradigms, such as Chain-of-Thought, with the aim of enhancing model performance on our dataset. However, dealing properly with negation may also require novel neural architectures.

## Limitations

The dataset contains a limited number of low-quality sentences, which are discussed in Section 3.2. Through manual evaluation, we find that over 96% of the sentences are considered understandable and grammatically correct by at least one human annotator and their prediction of whether the sentence is true or false matches the label in the dataset. Hence, the presence of low-quality sentences does not have a significant impact on the evaluation results. On the other side, a majority of sentences in the dataset are not plausible and unlikely to be spoken by English speakers. This feature provides a benefit by ensuring that the sentences are improbable to be found in the LLM training corpus, thereby it prevents models from relying solely on memorization to generate accurate responses.

All experiments were conducted by querying the models for the probability of True and False tokens. We did not explore more complex reasoning prompts, such as Chain of Thought. However, as explained in Section 7, we believe that models should be able to comprehend negation and provide accurate answers across diverse settings. Complex reasoning paradigms may not always be feasible in real-world applications, specially when models are used by non-NLP professionals.

Finally, the performance of models in our dataset is not solely determined by their capability to understand negation. Factors such as performance in question answering and prompting tasks, as well as their understanding of real-world knowledge, play a crucial role. However, models like Vicuna13B and Flan-T5-xxl showcase remarkable proficiency in correctly responding to affirmative sentences, indicating that their struggles primarily arise from the presence of negation. Additionally, we introduce a coherence metric that considers whether the model changes its prediction in the presence of negation, rather than solely focusing on the accuracy of the model's answer to the question.

## Ethics Statement

The dataset has been created through the English WordNet relations, so it reflects most of the "western" knowledge and might fall short in including concepts of non-English speaking communities. The generated triples from WordNet may include offensive or biased sentences. This can be caused by inherited biases from WordNet, or it can be caused unintentionally during the random sampling of synsets.

## Acknowledgements

We would like to thank Jeremy Barnes and Aritz Farwell for willingly offering themselves to conduct the dataset quality assessment experiments. Begoña Altuna is supported by the Basque Government postdoctoral grant POS 2022 2 0024. Iker García-Ferrero is supported by a doctoral grant from the Basque Government (PRE_2022_2_0208). This work has also been partially supported by HiTZ center and the Basque Government (Research group funding IT-1805-22). We also acknowledge the funding from the following projects:
(i) Antidote project funded by (PCI2020-120717-2) MCIN/AEI/10.13039/501100011033 and by "ERDF A way of making Europe"
(ii) MOTION (PID2020-112581GB-C22) supported by the Ministry of Science and Innovation of the Spanish Government
(iii) The Basque Project LoRea (UPV/EHU GIU21/044).
(iv) DeepKnowledge (PID2021-127777OB-C21) and ERDF A way of making Europe
(v) DeepR3 (TED2021-130295B-C31) and European Union NextGeneration EU/PRTR.

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

# A Extended zero-shot results

Apart from the models presented in Section 5, as anticipated, we have also tested the performance in the task of the following models: Koala [4], which is a LLaMA model that has been fine-tuned for dialogue; Open-Assistant (oasst-sft-1-pythia-12b) [5], which is a Pythia model fine-tuned on human generated assistant conversations; and INCITE [6] 7 billion foundation model along with the two models that have been further fine-tuned by the authors in the instruction-tuning paradigm and chat conversation. Table 8 shows the extended evaluation results.

# B Efficient inference and training

To facilitate inference of the models on a single GPU, we employed 8-bit quantization (Dettmers et al., 2022) for all of them. We conducted preliminary experiments with Vicuna 13 billion parameter model. Results are shown in Table 9. While the running cost of the models significantly decreases, we observed only minimal performance degradation in the quantified versions.

For the training process, we utilized Low-Rank Adaptation (LoRA) (Hu et al., 2022). This approach involves freezing the weights of the pre-

---

[4] https://bair.berkeley.edu/blog/2023/04/03/koala/

[5] https://huggingface.co/OpenAssistant/oasst-sft-1-pythia-12b

[6] https://www.together.xyz/blog/redpajama-7b

| Model name | Model Type | Coherence | | | Accuracy | | | | |
|---|---|---|---|---|---|---|---|---|---|
| | | | | | | W/o Distractor | | W/ Distractor | |
| | | All | W/o Distractor | W/ Distractor | All | Affirmation | Negation | Affirmation | Negation |
| Random | - | 0.5 | 0.9 | 0.8 | 50.0 | 50.2 | 50.1 | 50.0 | 49.9 |
| LLaMA7B | Foundation | 0.0 | 0.2 | 0.0 | 50.4 | 95.7 | 11.8 | 2.3 | 91.0 |
| LLaMA13B | Foundation | 0.0 | 0.2 | 0.3 | 50.1 | 85.8 | 12.2 | 10.6 | 90.2 |
| LLaMA30B | Foundation | 0.1 | 0.3 | 0.2 | 52.4 | 84.7 | 29.5 | 30.2 | 68.8 |
| LLaMA65B | Foundation | 0.0 | 0.0 | 0.0 | 50.3 | 96.3 | 3.1 | 1.3 | 99.3 |
| Vicuna7B | Dialogue | 0.0 | 0.2 | 0.0 | 52.4 | 18.1 | 96.9 | 98.9 | 0.3 |
| Vicuna13B | Dialogue | 0.2 | 8.8 | 0.6 | 57.8 | 83.1 | 85.1 | 78.0 | 2.6 |
| Koala7B | Dialogue | 0.0 | 0.0 | 0.0 | 50.1 | 5.4 | 97.2 | 99.5 | 0.3 |
| Koala13B | Dialogue | 0.0 | 0.7 | 0.0 | 52.8 | 20.4 | 97.1 | 98.8 | 0.3 |
| WizardLM7B | Instruction | 0.2 | 0.2 | 0.7 | 51.0 | 89.6 | 27.3 | 14.2 | 73.8 |
| WizardLM13B | Instruction | 0.0 | 4.3 | 0.3 | 57.4 | 68.7 | 86.2 | 87.5 | 2.9 |
| WizardLM30B | Instruction | 0.0 | 6.0 | 0.1 | 57.3 | 53.6 | 95.7 | 88.8 | 2.0 |
| WizardLM7B-uncensored | Instruction | 0.0 | 1.0 | 0.1 | 53.7 | 63.4 | 78.4 | 63.3 | 17.5 |
| WizardLM13B-uncensored | Instruction | 0.0 | 0.3 | 0.0 | 50.5 | 7.6 | 97.0 | 99.3 | 0.3 |
| WizardLM30B-uncensored | Instruction | 0.0 | 0.0 | 0.0 | 49.8 | 3.6 | **97.2** | **99.6** | 0.2 |
| Pythia12B | Foundation | 0.0 | 0.1 | 0.0 | 50.1 | 93.8 | 15.2 | 4.0 | 86.7 |
| oasst-pythia12B | DIalogue | 0.0 | 0.0 | 0.0 | 49.8 | 4.7 | 97.1 | 98.9 | 0.3 |
| Dolly12B | Instruction | 0.0 | 0.3 | 0.2 | 50.2 | 72.0 | 73.3 | 33.4 | 25.1 |
| T5-xxl | Foundation | 0.0 | 0.0 | 0.0 | 50.3 | **96.6** | 2.8 | 0.4 | **99.8** |
| Flan-T5-xxl | Instruction | **0.9** | **46.4** | **1.2** | **66.1** | 86.1 | 96.1 | 94.6 | 6.2 |
| Falcon7b | Foundation | 0.0 | 0.0 | 0.0 | 50.3 | **96.6** | 2.9 | 0.4 | 99.7 |
| Falcon7b-instruct | Instruction | 0.0 | 0.3 | 0.4 | 50.1 | 82.7 | 20.9 | 21.2 | 77.0 |
| Falcon40b | Foundation | 0.1 | 0.1 | 0.2 | 49.7 | 90.9 | 13.9 | 11.6 | 83.3 |
| Falcon40b-instruct | Instruction | 0.1 | 1.5 | 0.2 | 54.7 | 64.3 | 76.8 | 71.4 | 16.6 |
| INCITE7B-Base | Foundation | 0.1 | 0.3 | 0.1 | 50.3 | 82.6 | 17.0 | 16.4 | 84.5 |
| INCITE7B-Instruct | Instruction | 0.2 | 0.4 | 0.4 | 50.5 | 73.4 | 22.5 | 26.9 | 78.7 |
| INCITE7B-Chat | Dialogue | 0.1 | 0.3 | 0.2 | 50.0 | 19.8 | 89.4 | 88.0 | 6.0 |

Table 8: Zero-shot performance of various LLMs in our dataset. The best results are highlighted in **bold**, and scores that surpass the Random baseline accuracy are indicated with underline.

| Model name | Precision | Coherence | | | Accuracy | | | | |
|---|---|---|---|---|---|---|---|---|---|
| | | | | | | W/o Distractor | | W/ Distractor | |
| | | All | W/o Distractor | W/ Distractor | All | Affirmation | Negation | Affirmation | Negation |
| Vicuna13B | 8-Bits | 0.2 | 8.8 | 0.6 | **57.8** | 83.1 | 85.1 | **78.0** | 2.6 |
| Vicuna13B | Float16 | **0.4** | **10.1** | **1.1** | **57.8** | 84.4 | 85.8 | 74.7 | **3.2** |

Table 9: Zero-shot performance of Vicuna using 8-Bits quantification and the orifinal float16 weights. The best results are highlighted in **bold**, and scores that surpass the Random baseline are indicated with underline.

trained model and introducing trainable rank decomposition matrices into each layer. The frozen model weights are quantized into 8 bits, while the LoRA trainable weights remain in 16 bits (Dettmers et al., 2023). By adopting this efficient training paradigm, we were able to train LLMs with up to 13 billion parameters on a single GPU within a reasonable timeframe.

We perform all our experiments using a single NVIDIA A100 GPU with 80GB memory. The machine used has two AMD EPYC 7513 32-Core Processors and 1024GB of RAM.

## C   Dataset construction: selection of distractors

The automatic creation of false knowledge (distracting triples) on the basis of WordNet triples requires the use of distractors. In general, distrac-

| Pattern | W/o Distractor | | W/ Distractor | |
|---|---|---|---|---|
| | Affirmation | Negation | Affirmation | Negation |
| #01 | True | False | False | True |
| #02 | False | True | False | True |
| #03 | True | False | False | True |
| #04 | False | True | False | True |
| #05 | True | False | False | True |
| #06 | True | False | False | True |
| #07 | True | False | False | True |
| #08 | True | False | False | True |
| #09 | True | False | False | True |
| #10 | True | False | False | True |
| #11 | True | False | False | True |

Table 10: Truth-value of resulting sentences by pattern.

tors are randomly selected words that replace the word senses of a synset in a given triple, although the whole synset is replaced when using glosses in templates (Patterns #01 and #02). For each Word-Net triple, we use a single distracting triple except

for Patterns #02, #03 and #04, where we use two distracting triples obtained by using one distractor per synset. Apart from BLCs, for the selection of suitable distractors we consider the *lexicographer files* provided by WordNet, which are 45 syntactic category and logical groupings of word senses, and WordNet *Domains*, which consist of a hierarchy of 164 labels that characterize knowledge areas and to which each synset is connected. More concretely:

- Words (synsets in the case of Pattern #01 and #02 when using glosses) belonging to some BLCs cannot be distractors to ensure that selected words (synsets) are not too general.

- The combined lexicographer file and WordNet Domain annotation of any word sense of the given synset and of any synset where the distractor occurs (of the synset in the case of Pattern #01 and #02 when using glosses) have to be different.

In general, these restrictions ensure that the resulting false triples do not encode true knowledge. The probability of choosing a synset as distractor is directly proportional to the logarithm of its frequency.

For example, *wood* can be used as distractor of $expenditure_n^1$ because *wood* belongs to the lexicographer files *noun.substance* (nouns denoting cognitive processes and contents), *noun.group* (nouns denoting groupings of people or objects) and *noun.artifact* (nouns denoting man-made objects) while *expenditure* belongs to *noun.possession* (nouns denoting possession and transfer of possession) and *noun.act* (nouns denoting acts or actions). Therefore, we get the distracting triple $\langle ant, wood, income_n^1 \rangle$ from $\langle ant, expenditure_n^1, income_n^1 \rangle$. On the contrary, the word *registration* cannot be used as distractor of $expenditure_n^1$ as both words belong to the lexicographic file *noun.act* and the synsets $expenditure_n^2$ and $registration_n^1$ belong to the *economy* domain.

## D Dataset Description

In Table 10, we provide the truth-value of sentences that results by instantiating affirmative and negative templates using WordNet and distracting triples according to the pattern. In the following subsections, we describe each pattern and provide some examples that are used in Figures 3–8 to illustrate the instantiation of a sample of the templates. In all the templates described in Figures 3–8, alternative and optional expressions are enclosed respectively in square brackets and parentheses.

### D.1 Pattern #01: synonymy (gloss)

This pattern includes 21 templates stating semantic equivalence correspondences between a word and the gloss of a synset to which the word belongs. Since WordNet does not provide triples for synonymy relating two synsets, we get triples relating each synset to itself by reflexivity. For each resulting triple, templates are also instantiated using one distracting triple that is obtained by replacing the third component of each triple with a distractor (synset).

In Figure 3, we introduce a positive and a negative template and illustrate their instantiation using the synset $fligth_n^9$ with gloss "*a scheduled trip by plane between designated airports*" and the distractor $troop_n^1$ ("*a group of soldiers*").

### D.2 Pattern #02: antonymy (gloss)

This pattern includes 21 templates stating semantic equivalence correspondences between a word and the gloss of an antonym synset, where the word and the gloss are respectively taken from the second and third component of triples. Furthermore, for each WordNet antonymy triple templates are also instantiated using two distracting triples that are respectively obtained by replacing the second and third component with distractors (for the third one, the distractor is a synset).

In Figure 3, we introduce a positive and a negative template and illustrate their instantiation using the antonym synsets $brother_n^1$ and $sister_n^1$ ("*a female person who has the same parents as another person*") and the distractors *stream* and $fiction_n^1$ ("*a literary work based on the imagination and not necessarily on fact*").

### D.3 Pattern #03: synonymy

This pattern includes 24 templates stating semantic equivalence correspondences between words. Since WordNet does not provide triples for synonymy relating two synsets, we get triples relating each synset to itself by reflexivity. For each resulting triple, templates are also instantiated using two distracting triples that are respectively obtained by replacing the second and third component with distractors.

In Figure 4, we introduce a positive and a negative template and illustrate their instantiation using

the synonym words *path* and *route* and the distractors *engine* and *identity*.

### D.4 Pattern #04: antonymy

This pattern includes 24 templates stating semantic equivalence correspondences between words. For each WordNet antonymy triple, templates are also instantiated using two distracting triples that are respectively obtained by replacing the second and third component with distractors.

In Figure 5, we introduce a positive and a negative template and illustrate their instantiation using the antonym synsets $expenditure_n^1$ and $income_n^1$ and the distractors *wood* and *year*.

### D.5 Pattern #05: hypernymy

This pattern includes 24 templates stating semantic subsumption correspondences between words. For each WordNet hypernymy triple, templates are also instantiated using one distracting triple that is obtained by replacing the hyponym with a distractor.

In Figure 5, we introduce a positive and a negative template and illustrate their instantiation using the synset $auction_n^1$, which is hyponym of $sale_n^2$, and the distractor *breakdown*.

### D.6 Pattern #06: meronymy (part)

This pattern includes 16 templates stating semantic correspondences between parts and wholes. For each WordNet triple, templates are also instantiated using one distracting triple that is obtained by replacing the whole with a distractor.

In Figure 6, we introduce a positive and a negative template and illustrate their instantiation using the synset $week_n^3$, which is related by *part* with $month_n^1$, and the distractor *fence*.

### D.7 Pattern #07: meronymy (substance)

This pattern includes 15 templates stating semantic correspondences between substances and things. For each WordNet triple, templates are also instantiated using one distracting triple that is obtained by replacing the whole with a distractor.

In Figure 6, we introduce a positive and a negative template and illustrate their instantiation using the synset $sand_n^1$, which is related by *substance* with $beach_n^1$, and the distractor *decade*.

### D.8 Pattern #08: meronymy (member)

This pattern includes 17 templates stating semantic correspondences between members and groups.

For each WordNet triple, templates are also instantiated using one distracting triple that is obtained by replacing the group with a distractor.

In Figure 7, we introduce a positive and a negative template and illustrate their instantiation using the synset $voter_n^1$, which is related by *member* with $electorate_n^1$, and the distractor *sport*.

### D.9 Pattern #09: semantic role (agent)

This pattern includes 2 templates stating semantic correspondences between agents and events. For each WordNet triple, templates are also instantiated using one distracting triple that is obtained by replacing the agent with a distractor.

In Figure 7, we introduce a positive and a negative template and illustrate their instantiation using the synset $rule_n^1$, which is related by *agent* with $governor_n^1$, and the distractor *hole*.

### D.10 Pattern #10: semantic role (instrument)

This pattern includes 7 templates stating semantic correspondences between instruments and events. For each WordNet triple, templates are also instantiated using one distracting triple that is obtained by replacing the event with a distractor.

In Figure 8, we introduce a positive and a negative template and illustrate their instantiation using the synset $telephone_n^1$, which is related by *instrument* with $call_v^3$, and the distractor *lay*.

### D.11 Pattern #11: semantic role (result)

This pattern includes 27 templates stating semantic correspondences between results and events. For each WordNet triple, templates are also instantiated using one distracting triple that is obtained by replacing the event with a distractor.

In Figure 8, we introduce a positive and a negative template and illustrate their instantiation using the synset $response_n^1$, which is related by *result* with $answer_v^1$, and the distractor *dress*.

**Pattern #01**: synonymy (gloss)

Affirmative template:

$$\text{A/An } \langle word \rangle \text{ is (commonly) } \langle gloss \rangle.$$

Sentences:

| | |
|---|---|
| A flight is commonly a scheduled trip by plane between designated airports. | True |
| A flight is a scheduled trip by plane between designated airports. | True |
| A flight is commonly a group of soldiers. | False |
| A flight is a group of soldiers. | False |

Negative template (verbal, analytic and clausal):

$$\text{A/An } \langle word \rangle \text{ is not } \langle gloss \rangle.$$

Sentences:

| | |
|---|---|
| A flight is not a group of soldiers. | True |
| A flight is not a scheduled trip by plane between designated airports. | False |

**Pattern #02**: antonymy (gloss)

Affirmative template:

$$\langle word \rangle \text{ (commonly) [ stands for | refers to ] } \langle gloss \rangle.$$

Sentences:

| | |
|---|---|
| Brother commonly stands for a female person who has the same parents as another person. | False |
| Brother commonly refers to a female person who has the same parents as another person. | False |
| Brother stands for a female person who has the same parents as another person. | False |
| Brother refers to a female person who has the same parents as another person. | False |
| Stream commonly stands for a female person who has the same parents as another person. | False |
| Stream commonly refers to a female person who has the same parents as another person. | False |
| Stream stands for a female person who has the same parents as another person. | False |
| Stream refers to a female person who has the same parents as another person. | False |
| Brother commonly stands for a literary work based on the imagination and not necessarily on fact. | False |
| Brother commonly refers to a literary work based on the imagination and not necessarily on fact. | False |
| Brother stands for a literary work based on the imagination and not necessarily on fact. | False |
| Brother refers to a literary work based on the imagination and not necessarily on fact. | False |

Negative template (synthetic and subclausal):

$$\text{A/An } \langle word \rangle \text{ is never } \langle gloss \rangle.$$

Sentences:

| | |
|---|---|
| A brother is never a female person who has the same parents as another person. | True |
| A stream is never a female person who has the same parents as another person. | True |
| A brother is never a literary work based on the imagination and not necessarily on fact. | True |

Figure 3: Description of Patterns #01 and #02.

**Pattern #03**: synonymy

Affirmative template:

$\langle noun_1$+(e)s$\rangle$ and $\langle noun_2$+(e)s$\rangle$ [ are | may be ] always different.

Sentences:

| | |
|---|---|
| Path and route are always different. | False |
| Path and route may be always different. | False |
| Engine and route are always different. | True |
| Engine and route may be always different. | True |
| Path and identity are always different. | True |
| Path and identity may be always different. | True |

Negative template (verbal, analytic and subclausal):

$\langle noun_1$+(e)s$\rangle$ and $\langle noun_2$+(e)s$\rangle$ [ are not | may not be ] synonyms in any context.

Sentences:

| | |
|---|---|
| Path and route are not synonyms in any context. | False |
| Path and route may not be synonyms in any context. | False |
| Engine and route are not synonyms in any context. | True |
| Engine and route may not be synonyms in any context. | True |
| Path and identity are not synonyms in any context. | True |
| Path and identity may not be synonyms in any context. | True |

Figure 4: Description of Pattern #03.

**Pattern #04**: antonymy

Affirmative template:

$\langle noun_1+(e)s \rangle$ and $\langle noun_2+(e)s \rangle$ [ are | may be ] synonyms (in certain contexts).

Sentences:

| | |
|---|---|
| Expenditure and income are synonyms in certain contexts. | False |
| Expenditure and income may be synonyms in certain contexts. | False |
| Expenditure and income are synonyms. | False |
| Expenditure and income may be synonyms. | False |
| Expenditure and year are synonyms in certain contexts. | False |
| Expenditure and year may be synonyms in certain contexts. | False |
| Expenditure and year are synonyms. | False |
| Expenditure and year may be synonyms. | False |
| Wood and income are synonyms in certain contexts. | False |
| Wood and income may be synonyms in certain contexts. | False |
| Wood and income are synonyms. | False |
| Wood and income may be synonyms. | False |

Negative template (analytic and subclausal):

$\langle noun_1+(e)s \rangle$ and $\langle noun_2+(e)s \rangle$ [ are | may be ] the same thing in no context.

Sentences:

| | |
|---|---|
| Expenditure and income are the same thing in no context. | True |
| Expenditure and income may be the same thing in no context. | True |
| Expenditure and year are the same thing in no context. | True |
| Expenditure and year may be the same thing in no context. | True |
| Wood and income are the same thing in no context. | True |
| Wood and income may be the same thing in no context. | True |

**Pattern #05**: hypernymy

Affirmative template:

A/An $\langle hyponym \rangle$ [ is | may be ] a/an $\langle hypernym \rangle$ in certain contexts.

Sentences:

| | |
|---|---|
| An auction is a sale in certain contexts. | True |
| An auction may be a sale in certain contexts. | True |
| A breakdwon is a sale in certain contexts. | False |
| A breakdown may be a sale in certain contexts. | False |

Negative template (synthetic and subclausal):

A/An $\langle hyponym \rangle$ is never a/an $\langle hypernym \rangle$.

Sentences:

| | |
|---|---|
| An auction is never a sale. | False |
| A breakdown is never a sale. | True |

Figure 5: Description of Patterns #04 and #05.

**Pattern #06**: meronymy (part)

Affirmative template:

> A/An $\langle part \rangle$ [ is commonly | may be ] part of a/an $\langle whole \rangle$.

Sentences:

| | |
|---|---|
| A week is commonly part of a month. | True |
| A week may be part of a month. | True |
| A week is commonly part of a fence. | False |
| A week may be part of a fence. | False |

Negative template (synthetic and subclausal):

> A/An $\langle part \rangle$ is never part of a/an $\langle whole \rangle$.

Sentences:

| | |
|---|---|
| A week is never part of a month. | False |
| A week is never part of a fence. | True |

**Pattern #07**: meronymy (substance)

Affirmative template:

> $\langle thing + (e)s \rangle$ [ are commonly | may be ] made of $\langle substance \rangle$.

Sentences:

| | |
|---|---|
| Beaches are commonly made of sand. | True |
| Beaches may be made of sand. | True |
| Decades are commonly made of sand. | False |
| Decades may be made of sand. | False |

Negative template (Analytic and subclausal):

> In no context $\langle thing + (e)s \rangle$ [ are |may be ] made of $\langle substance \rangle$.

Sentences:

| | |
|---|---|
| In no context beaches are made of sand. | False |
| In no context decades are made of sand. | True |

Figure 6: Description of Patterns #06 and #07.

**Pattern #08**: meronymy (member)

Affirmative template:

$$\langle member + (e)s \rangle \text{ [ are | may be ] members of } \langle group + (e)s \rangle.$$

Sentences:

| | |
|---|---|
| Voters are members of electorates. | True |
| Voters may be members of electorates. | True |
| Voters are members of sports. | Falses |
| Voters may be members of sports. | False |

Negative template (verbal, analytic and clausal):

$$\langle member + (e)s \rangle \text{ [ are not | may not be ] members of } \langle group + (e)s \rangle \text{ in any context.}$$

Sentences:

| | |
|---|---|
| Voters are not members of electorates in any context. | False |
| Voters may not be members of electorates in any context. | False |
| Voters are not members of sports in any context. | True |
| Voters may not be members of sports in any context. | True |

**Pattern #09**: semantic role (agent)

Affirmative template:

$$\langle event + ing \rangle \text{ is commonly done by } \langle agent + (e)s \rangle.$$

Sentences:

| | |
|---|---|
| Ruling is commonly done by governors. | True |
| Ruling is commonly done by holes. | False |

Negative template (Verbal, analytic and clausal):

$$\langle event + ing \rangle \text{ is not commonly done by } \langle agent + (e)s \rangle.$$

Sentences:

| | |
|---|---|
| Ruling is not commonly done by governors. | False |
| Ruling is not commonly done by holes. | True |

Figure 7: Description of Patterns #08 and #09.

**Pattern #10**: semantic role (instrument)

Affirmative template:

A/An ⟨*instrument*⟩ [ is commonly | may be ] [ used | needed ] for ⟨*event* + *ing*⟩.

Sentences:

| | |
|---|---|
| A telephone is commonly used for calling. | True |
| A telephone is commonly needed for calling. | True |
| A telephone may be used for calling. | True |
| A telephone may be needed for calling. | True |
| A telephone is commonly used for laying. | False |
| A telephone is commonly needed for laying. | False |
| A telephone may be used for laying. | False |
| A telephone may be needed for laying. | False |

Negative template (Synthetic and subclausal):

A/An ⟨*instrument*⟩ should never be [ used | needed ] for ⟨*event* + *ing*⟩.

Sentences:

| | |
|---|---|
| A telephone should never be used for calling. | False |
| A telephone should never be used for laying. | True |

**Pattern #11**: semantic role (result)

Affirmative template:

⟨*event* + *ing*⟩ [ commonly leads | may lead ] to a/an ⟨*result*⟩.

Sentences:

| | |
|---|---|
| Answering commonly leads to a response. | True |
| Answering may lead to a response. | True |
| Dressing commonly leads to a response. | False |
| Dressing may lead to a response. | False |

Negative template (analytic and subclausal):

⟨*event* + *ing*⟩ [ leads | may lead ] to a/an ⟨*result*⟩ in no context.

Sentences:

| | |
|---|---|
| Answering leads to a response in no context. | False |
| Answering may lead to a response in no context. | False |
| Answering leads to a response in no context. | True |
| Answering may lead to a response in no context. | True |

Figure 8: Description of Patterns #10 and #11.