# OpenReview forum: "This is not a Dataset: A Large Negation Benchmark to Challenge Large Language Models"
_EMNLP/2023/Conference — EMNLP 2023 Main_

### Official Review · Reviewer_rrGn · 2023-08-03

**Soundness:** 5

**Excitement:**

4: Strong: This paper deepens the understanding of some phenomenon or lowers the barriers to an existing research direction.

**Paper Topic And Main Contributions:**

The paper investigates the linguistic phenomenon of negation within the context of LLMs. In particular, the authors use linked lexical resources (WordNet) to create a large data set of sentences containing negation and then employ this data to explore the ability of LLMs to coherently understand negation. The authors also explore ways to improve that understanding.

This is an excellent linguistic study on the topic of negation and its handling by LLMs. I was really happy to see state-of-the-art LLM technology explored through the magnifying glass of sound linguistic experiments. We must better understand  how LLMs handle certain linguistic phenomena and for this, we need well-grounded linguistic analysis.

The authors employ well-motivated and exemplary linguistic approach to create the data set. In itself, this is already a great contribution. Yet, the authors go two steps further by i) exploring the performance of state-of-the-art LLMs of different sizes and pursposes and ii) trying standard approaches such as fine-tuning and retraining to improve the LLM performance on negation. Finally, despite achieving improved results on the data set, the authors do perform additional critical experiments, only to show that this improvement was not the result of LLMs "understanding" negation but rather of the increased number of negation patterns in the data set.

The paper is a result of a tremendous amount of sound work and experiments, looking at and analysing the issue at hand from various view points. The submission is examplary for the quality an EMNLP paper should have.

**Questions For The Authors:**

I am not sure I completely understand the definition of overall coherence. Could you please elaborate on that in your response and make the definition a bit more clear in the camera-ready version should the paper be accepted?

Can you speculate a bit on LLM performance for languages which employ negative concord to express negation, i.e. using multiple negation tokens to express a single semantic negation? For example, French and Spanish have double negation and some Slavic languages easily have 4 negation tokens in a single sentence. Would this repetative pattern lead to better LLM performance or even allow LLMs to generalize and understand negation better than in English? This is not directly connected to the paper but I would be interested in a short answer whether you have considered that phenomenon or are aware of any work in that direction.

**Reasons To Accept:**

- The negation data set is a valuable resource for further research. It is the result of sound linguistic reasoning and is representative of various aspects of negation
- Good qualitative analysis of the data set performed by human annotators
- Vast evaluation of LLMs of different sizes and types, with well-founded explanations for the observed performance and behaviour of the models
- The fine-tuning experiment with Vicuna and each pattern separately (results presented in Figure 2) which clearly shows that "While exposure to negation may contribute to achieving favorable results within a specific dataset, it does not lead to a generalization on negation by the models"

The biggest plus of the paper is the bringing of LLMs and linguistics together. I think such methods are needed more and more, if LLMs are to become more successful.

**Reasons To Reject:**

None really. The only thing I can slightly complain about is the relatively low number of examples evaluated by humans. I would have liked to see may be a thousand examples which were not randomly selected from the overall data set but randomly selected from each pattern. The number of examples from the various patterns should reflect their distribution in the data set. This would make the qualitative evaluation of the data even stronger.

**Reproducibility:**

5: Could easily reproduce the results.

**Reviewer Confidence:**

5: Positive that my evaluation is correct. I read the paper very carefully and I am very familiar with related work.

**Typos Grammar Style And Presentation Improvements:**

I think that you should have either mentioned all LLMs involved in the experiments or at least summarized the extended zero-shot results in the main paper and not in Appendix A. Currenlty, this appendix presents new results which is against submission policy. Hence I am not considering the reported results in Appendix A for my review. Also, major technical setup details from Appendix B should also be in the main paper.

---

> ### Author Rebuttal · Authors · 2023-08-28
>
> Dear reviewer,
>
> Thank you for taking the time to thoroughly assess our contributions. Your questions and suggestions are addressed below. We also include some additional remarks.
>
> In what refers to coherence and overall coherence, take Figure 1 as an example: an LLM might correctly label the sentence "Bills are commonly part of birds" as True. Consequently, considering the negated "Bills are never part of birds", the model should predict the opposite label, False. In this scenario, we would consider the answer coherent. However, if the model predicts the same label for both the affirmative and negated sentences, we would consider the answer not coherent. Conversely, if the model lacks the correct real-world knowledge and labeled "Bills are commonly part of birds" as False, but labeled the negated sentence as True, we would still consider the answer as coherent. This is why we claim that coherence is a metric that seeks to distinguish the model's real-world knowledge from its comprehension of negation.
>
> We compute two coherence scores: one for the sentences without distractors ("Bills are commonly part of birds" and "Bills are never part of birds") and another for the sentences with distractors ("Bills are commonly part of human bodies." and "Bills are never part of human bodies."). For overall coherence, we expect the answers for the statements, both with and without distractors, to be not only coherent individually but also all of them must be correctly or all incorrectly classified. Referring to the example in Figure 1, we would deem the set of statements as overall coherent if the sentences with and without distractors are coherent and the model predicts opposite labels for "Bills are commonly part of birds" versus "Bills are commonly part of human bodies," and for "Bills are never part of birds" versus "Bills are never part of human bodies." In the case of antonymy relations (Patterns #2 and #4), both the non-distracted and distracted sentences carry the same label, so we evaluate the overall coherence accordingly.
>
> It's important to note that, for the sake of simplicity, the example only contains two sentences, but coherence is actually determined at the triple level. Triples can comprise between 2 to 27 templates, and for a triple to be deemed coherent, responses to all the templates within it must be coherent. Therefore, this is a very challenging metric. We will clarify the definition of coherence and overall coherence in the paper.
>
> Analyzing the impact of different types of negation was one of the main goals of this paper. As you mention, English does not have negative concord, but other languages do. It is effectively out of the scope of this paper, but we are already working on a similar experiment in other languages. For this, we are currently working on the template generation and we have not tested the LLMs yet. Nonetheless, analyzing the results for English, in which the negation was placed in different parts of the sentence and was affecting different elements in it, e.g. some negations are verbal, some negate the subject, etc., we have not noticed significant differences in the performance of LLMs (see Table 7). As a consequence, our intuition is that concord negatives will be understood in the same terms as for the sentences that contain single negation cues. Nonetheless, we expect that our experiment in other languages will shed some light on the issue.
>
> Regarding the number of sentences revised by the human assessor, we will clarify in the paper that Human Evaluation addresses the validation of the generation process and the different templates used.

---

### Official Review · Reviewer_jGqB · 2023-08-04

**Typos Grammar Style And Presentation Improvements:** Line 611
**Soundness:** 3

**Excitement:**

4: Strong: This paper deepens the understanding of some phenomenon or lowers the barriers to an existing research direction.

**Paper Topic And Main Contributions:**

The focus of this paper is to investigate whether LLMs are capable of understanding negation. The author presents a dataset of 400K examples having True/False labels where 2/3 of the dataset contains negation in various forms. With the zero-shot and fine-tuning experiments, the authors find that negation is still a challenge in LLMs.

**Questions For The Authors:**

What are the chances of making semantically incorrect sentences if you use the distractors? If so, should those sentences be included in the experiments?

**Reasons To Accept:**

The paper presents a quite large corpus containing (mostly) negation and affirmation examples which can be useful in future research. The authors perform a thorough analysis with zero-shot and fine-tuning approaches to evaluate performance of the recent LLMs when negation is present.

**Reasons To Reject:**

Issues and improvement areas:

1) The construction of the dataset (Section 3) is little ambiguous. It would be nice to have sufficient examples for the different patterns described in Table 2.

2) Only 220 examples are assessed out of 400K examples! Further, Table 3 should have a elaborate caption detailing the meaning of the last two columns.

3) As this paper works with the LLMs, authors should check the in-context leaning (i.e., few-shot prompting) performance as well.

4) The evaluation metric "Coherence" is ambiguous. Explaining how it is calculated would be helpful.

5) The core insights from Table 4 seem missing. For example, why does LLaMA65B achieve 99.3% accuracy with negation (w/ distractor) but only 1.3% accuracy on the affirmation?




**Reproducibility:**

4: Could mostly reproduce the results, but there may be some variation because of sample variance or minor variations in their interpretation of the protocol or method.

**Reviewer Confidence:**

3: Pretty sure, but there's a chance I missed something. Although I have a good feel for this area in general, I did not carefully check the paper's details, e.g., the math, experimental design, or novelty.

---

> ### Author Rebuttal · Authors · 2023-08-28
>
> Dear reviewer,
>
> Thank you for taking the time to thoroughly assess our contributions. Your questions are addressed below. We also include some additional remarks.
>
> The chances of making semantically incorrect sentences are very low. On one hand, WordNet only contains true knowledge which accounts for 50% of the total dataset. The random selection of distractors for building counterfactual sentences could lead to semantically incorrect assertions that encode true or false knowledge when the opposite should be expected. Thus, we have defined a set of restrictions for distractor selection to avoid these situations as it is described in Appendix C. In the human evaluation experiment, we have checked that over 96% of the sentences are considered understandable, grammatically correct by at least one human annotator and that their prediction of whether the sentence is true or false matches the label in the dataset (see Table 3 in the paper).
>
> Regarding the number of sentences revised by the human assessors. We will clarify in the paper that Human Evaluation addresses the validation of the generation process and the different templates used.
>
> We would like to address some of the concerns about our paper:
>
> - Due to the 8 page limit, an in-depth description and examples of each template described in Section 3 is included in Appendix D.
>
> - We fully agree about the remark about testing in-context learning, as well as other prompt paradigms such as “chain of thought”. We plan to continue experimenting with our dataset and we will definitely test in-context learning in the future. However, as we explain in the Limitations section, we believe that models should be able to comprehend negation and provide accurate answers across diverse settings.
>
> - The scores for LLaMA65B correspond to a model that labels every statement, negated or not, as “True”. It achieves near 100% accuracy in the set of statements that are gold labeled as “True” and near 0% accuracy in the set of sentences gold labeled as “False”. We further elaborate on this behavior at the beginning of Section 5.
>
> - We will rewrite it to better explain the coherence metric. You can find further explanations of the metric in the answer to Reviewer rrGn.

---

### Official Review · Reviewer_rCcw · 2023-08-11

**Soundness:** 4

**Excitement:**

4: Strong: This paper deepens the understanding of some phenomenon or lowers the barriers to an existing research direction.

**Paper Topic And Main Contributions:**

Topic : A dataset to benchmark negation understanding in language models.

Contributions:
A. Synthetically generated largest probing negation dataset.
B. Evaluations and findings of negation understanding in  Large Language Models (LLMs) in zero-shot setting and after fine tuning. The paper asserts that LLMs lack the capability to comprehend negation.


**Questions For The Authors:**

A. While determining the veracity of a sentence as true or false is a step forward, it leaves open the inquiry into whether models continue to depend on surface-level cues for their responses. A more effective approach involves crafting prompts that predict the concealed token, wherein the predicted token adheres to the same distribution in cases of affirmation, and conversely, originates from a distinct distribution when negation is involved.

B. As stated "Current LLMs are typically trained using next token or mask token prediction objectives, which have proven effective for various NLP tasks. However, it remains uncertain whether this training paradigm adequately models negation".  Have you attempted fine-tuning the model using your dataset as a means of substantiating this assertion?

**Reasons To Accept:**

A. To proficiently enhance negation, a dataset containing affirmative and negative sentences that align with the input criteria of Language Models (LLMs) is essential. Unfortunately, a publicly accessible dataset possessing these specific attributes is currently absent. The paper endeavors to address this gap.

B. The dataset's composition and its inherent contents are comprehensively elucidated.

C. The conducted analysis of the dataset satisfactorily demonstrates the alignment with the paper's hypothesis regarding the inadequate comprehension of negation by LLMs.

**Reasons To Reject:**

A. While the dataset is comprehensive, it falls short of being labeled a gold standard due to the limited proportion evaluated by human assessors.

B. Incorporating various additional approaches to assess the model's grasp of negation would have been more advantageous. This would have enhanced the credibility of the dataset.

**Reproducibility:**

5: Could easily reproduce the results.

**Reviewer Confidence:**

5: Positive that my evaluation is correct. I read the paper very carefully and I am very familiar with related work.

---

> ### Author Rebuttal · Authors · 2023-08-28
>
> Dear reviewer,
>
> Thank you for taking the time to thoroughly assess our contributions. Your questions are addressed below. We also include some additional remarks.
>
> Regarding question A, in this paper, we are primarily interested in evaluating the recent wave of instruction/chat LLMs. These models are expected to be able to answer questions, therefore, we defined our prompts as a question answering task. We find your suggestion very interesting, for it allows for a more thorough linguistic analysis of the results, and we will try to explore it in the near future. As our dataset will be publicly available, alternative paradigms can also be explored. Regarding the comment about superficial cues, we believe that our experiments prove that, in fact, what you mention is what happens in both the zero-shot and fine-tuning experiments. This is especially obvious in the comparison of accuracy in the Synonymy and Antonymy patterns in Table 5. We further elaborate on this topic in Section 5.
>
> Regarding question B. We believe that it is important to study how LLMs can be trained to model negation effectively and that our dataset can help to study this phenomena. Nevertheless, we will rephrase this sentence, as we might have overstepped in this assertion.
>
> Regarding the number of sentences revised by the human assessors, we will clarify in the paper that Human Evaluation addresses the validation of the generation process and the different templates used, that is to say, whether the sentences in the dataset are grammatical and that overall represent true and false knowledge as expected.
>
> In what refers to the grasping of negation by models, it is true that there might be other approaches, but these would have forced us to vary drastically the configuration of the experiment. Still, we consider them worth the effort for future work as negation seems still an unsolved issue in NLU. Nonetheless, following a linguistic intuition, we have tried to see whether different types of negations affecting different parts of the sentence and placed in different positions within the sentence affect the performance of the LLMs, which has allowed us to assess the grasping of negation in a somewhat wide range of scenarios.

---

### Meta-Review · Area_Chair_Emvd · 2023-09-15

**Recommendation:** 5

**Metareview:**

The focus of this paper is to investigate whether LLMs are capable of understanding negation. The authors presents a dataset containing utterances with various forms of negation. It is found that negation is still a challenge for LLMs.

The reviewers found that this paper was an excellent linguistic study on the topic of negation and its handling by LLMs and that the provided resource would be very useful for future research. It also fits the EMNLP theme track particularly well. The main criticism identified by the reviewers lies in the somewhat small part of the corpus that was evaluated by human assessors.

For future work, it would also be interesting to see versions of this benchmark in different languages, especially languages that use different negation patterns (e.g. negative concord).

---

### Decision · Program_Chairs · 2023-10-07

**Decision:**

Accept-Main

**Comment:**

The focus of this paper is to investigate whether LLMs are capable of understanding negation. The authors presents a dataset containing utterances with various forms of negation. It is found that negation is still a challenge for LLMs.

The reviewers found that this paper was an excellent linguistic study on the topic of negation and its handling by LLMs and that the provided resource would be very useful for future research. It also fits the EMNLP theme track particularly well. The main criticism identified by the reviewers lies in the somewhat small part of the corpus that was evaluated by human assessors.

For future work, it would also be interesting to see versions of this benchmark in different languages, especially languages that use different negation patterns (e.g. negative concord).